# Revelation for the Influence Mechanism of Long-Chain Fatty Acid Ethyl Esters on the Baijiu Quality by Multicomponent Chemometrics Combined with Modern Flavor Sensomics

**DOI:** 10.3390/foods12061267

**Published:** 2023-03-16

**Authors:** Yashuai Wu, Hao Chen, He Huang, Fangyuan Chen, Jiaxin Hong, Dongrui Zhao, Chunsheng Zhang, Zhigang Zhao, Shimin Wang, Ran Ao, Baoguo Sun

**Affiliations:** 1Key Laboratory of Brewing Molecular Engineering of China Light Industry, Beijing Technology and Business University, Beijing 100048, China; 2Beijing Laboratory of Food Quality and Safety, Beijing Technology and Business University, Beijing 100048, China; 3College of Insurance, University of International Business and Economics, Beijing 100105, China; 4Department of Nutrition and Health, China Agriculture University, Beijing 100193, China; 5Chengde Qianlongzui Distillery Company, Chengde 067400, China

**Keywords:** Baijiu, long-chain fatty acid ethyl ester, softness, aroma characteristics, modern flavor sensomics

## Abstract

Long-chain fatty acid ethyl ester (LCFAEEs) is colorless and has a weak wax and cream aroma. It can be used as an intermediate for the synthesis of emulsifiers, and stabilizers and be applied in the production of flavor essence. It is also an important trace component in Baijiu and is attributed to making a contribution to the quality of Baijiu, but its distribution in Baijiu has not been clear, and its influence mechanisms on Baijiu quality have not been systematically studied. Therefore, the distribution of LCFAEEs for Baijiu in different years (2014, 2015, 2018, and 2022), different grades (premium, excellent, and level 1; note: here Baijiu grade classification was based on Chinese standard (GB/T 10781) and enterprise classification standard), and different sun exposure times (0, 6, 12, 20, 30, and 50 days) was uncovered. Thus, in this study, the effect of LCFAEEs on the quality of Baijiu was comprehensively and objectively proven by combining modern flavor sensomics and multicomponent chemometrics. The results showed that with the increase in Baijiu storage time, the concentration of LCFAEEs increased significantly in Baijiu (4.38–196.95 mg/L, *p* < 0.05). The concentration of LCFAEEs in level 1 Baijiu was significantly higher than that in excellent and premium Baijiu (the concentration ranges of ET, EP, EO, E9, E912, and E91215 were: 0.27–2.31 mg/L, 0.75–47.41 mg/L, 0.93–1.80 mg/L, 0.98–12.87 mg/L, 1.01–27.08 mg/L, and 1.00–1.75 mg/L, respectively, *p* < 0.05). With the increase in sun exposure time, the concentration of LCFAEEs in the Baijiu first increased significantly and then decreased significantly (4.38–5.95 mg/L, *p* < 0.05). As the flavor sensomics showed, the concentrations of LCFAEEs in Baijiu bodies were significantly correlated with the Baijiu taste sense (inlet taste, aroma sensation in the mouth), as well as with the evaluation after drinking (maintaining taste) (*p* < 0.05, *r* > 0.7). Based on the above, LCFAEEs are critical factors for Baijiu flavor thus, it is essential to explore a suitable concentration of LCFAEEs in Baijiu to make Baijiu’s quality more ideal.

## 1. Introduction

Distilled liquor refers to liquor that is made from grains, fruits, etc. and obtained through fermentation, distillation, aging, and blending. Its alcohol content is mostly between 38% and 65% (standard alcohol content, calculated by volume fraction) [1]. The world’s top six distilled liquors include French brandy, Russian vodka, English whisky, Cuban rum, Dutch gin, and Chinese Baijiu [1,2]. In contrast, Chinese Baijiu (hereinafter referred to as Baijiu) has its own distinctive characteristics in terms of raw materials, production technology, and style characteristics that are more complicated than those of other distilled liquors. Specifically, Baijiu has a rich aroma as well as a soft, sweet, mellow, dry, and refreshing taste. Thus, it is deeply loved by the Chinese [3,4,5,6,7]. Undoubtedly, Baijiu plays an important role in the world’s distilled liquor industry. At present, the annual output of distilled liquor in the world is close to 30 million tons, of which the output of Baijiu is the highest. According to statistics, the annual output of Baijiu enterprises above the designated size was about 8 million tons, accounting for more than 30% of the world’s total, which was about the total output of brandy, whisky, and vodka. Thus, China is the world’s largest distilled liquor market and the country with the fastest growing demand for it. Its substantial population and booming economy, as well as changing tastes and preferences, all point to sustained growth in the demand for distilled spirits in the years to come. In detail, China’s consumption of Baijiu ranks first in the world liquor industry, and its consumption accounts for one third of the total global liquor consumption [8,9,10,11].

Not to make your throat spicy and your head dizzy is the consistent world evaluation of high-quality distilled liquor, which predominantly describes the taste dimension and the core of Baijiu quality [12,13,14,15]. At present, the research on the aroma of Baijiu in China is relatively complete, while the research on taste is less. The ongoing research shows that the trace components in Baijiu play a decisive role in the flavor quality of Baijiu, specifically in the aroma characteristics of Baijiu. Long-chain fatty acid ethyl esters (LCFAEEs) are ubiquitous in Baijiu, and the origin of LCFAEEs is currently thought to originate from three pathways (i.e., raw material, microbial synthetic, and chemical esterification) [16,17,18,19]. It has been reported that the LCFAEEs may have an impact on the “non throat spicy” of Baijiu [20,21,22]. However, the distribution of LCFAEEs in Baijiu has not been clearly defined, and its effect on the quality of Baijiu has not been systematically studied.

Baijiu has been developing for thousands of years and has gradually formed mature production techniques (Figure 1). The production process of Baijiu is elaborate and predominantly consists of five steps. The basis for producing perfect Baijiu is having good-quality raw materials. In the history of brewed Baijiu in our country, Wugu grains (i.e., rice, wheat, soybeans, corn, tubers), but also potato species containing both starch and sugar, can be used in the fermentation of Baijiu. “Qu” is the “bone” of Baijiu, making “Qu” a pivotal link in brewing Baijiu. The fermentation process is used to transform the sugars contained in the raw materials into alcohol, and the Baijiu with higher quality is mostly fermented by solid-state fermentation. After that, the raw Baijiu is collected and stored. Of note, the flavor of raw Baijiu that has just been steamed is spicy and stimulating. It requires a process of aging to turn the raw Baijiu with an unsatisfactory taste into a mellow and soft Baijiu [23,24,25,26]. The aging process is carried out during the storage of Baijiu. During the aging process, the strong volatile and pungent trace components (such as volatile sulfides and aldehydes) in Baijiu will naturally volatilize, and their concentrations will decrease. Meanwhile, the trace components will be associated, and the concentrations of alcohol, aldehyde, acid, ester, etc. will further coordinate and balance, thus enhancing the quality of Baijiu and lowering its pungency. Generally, the aging process of Baijiu is predominantly dark, and the storage time is up to 2–5 years [27,28,29,30]. In order to shorten the aging time of Baijiu, the distillery has innovated a new aging technique based on practical experience. The new aging technique (marked in Figure 1) refers to exposing the raw Baijiu to the sun during the aging process. Through practical experience, the softness of sun-exposed Baijiu improved. It may be that transformations occurred between the trace components in Baijiu (i.e., alcohols, aldehydes, acids, and esters), but the mechanisms behind the trace component changes were unknown. Whether the concentration of LCFAEEs changed, especially in Baijiu, and influenced the flavor quality of Baijiu deserve further exploration.

At present, although various advanced detection instruments emerge endlessly and develop rapidly, they still cannot provide reliable data for the sensory evaluation of Baijiu. For Baijiu, the trace components contained in it are complicated and diverse, and the instrument cannot carry out sensory evaluation on Baijiu, which is also the magic of Chinese Baijiu. Nowadays, sensory evaluations of Baijiu are primarily focused on flavor rounds constructed from Baijiu flavor attributes. However, the sensory evaluation of Baijiu in actual production mainly uses the four-dimensional method of color, aroma, flavor, and feeling, which corresponds better to actual drinking behavior. For this reason, this study combined the two measures to evaluate the sensory quality of Baijiu samples.

Based on the above, the purpose of this study was to investigate the relationship between LCFAEEs and the quality of Baijiu. The findings of this research will enable us to gain a better understanding of the effect these LCFAEEs have on the quality of Baijiu and any possible implications for its production. Thus, this research selected the Baijiu with the above-mentioned aging production process, representative year, and representative grade as the research object. According to current research reports, the most common LCFAEEs (i.e., ethyl tetradecanoate, ethyl palmitate, ethyl octadecenoate, ethyl 9-octadecenoate, ethyl 9,12-octadecadienoate, and ethyl 9,12,15-octadecatrienoate) in Baijiu, were selected as the target components. Specifically, a method for identification and analysis of LCFAEEs in Baijiu was established to clarify the distribution of LCFAEEs in Baijiu for different years, grades, and sun exposure times. This research provides additional theoretical references for the future Baijiu industry to further enhance production technology and provides support for facilitating the modernization of the Baijiu industry.

## 2. Materials and Methods

### 2.1. Standard Information

The purity and source of standard information are shown in Table 1 below.

### 2.2. Baijiu Sample Information

The sample information for Baijiu is shown in Table 2 below. Baijiu samples were taken from a distillery: premium, excellent, and level 1 (note: here Baijiu grade classification was based on the Chinese standard (GB/T 10781) and enterprise classification standard) Baijiu produced in 2014; premium Baijiu produced in 2015; level 1 Baijiu produced in 2018; and level 1 Baijiu produced in 2022 after sun exposure for 0, 6, 12, 20, 30, and 50 days (timing using a watch, 24 h system). Note: The parallel Baijiu samples used in this experiment were taken from different storage tanks to ensure the practicality of the experimental data. Note that the sources of Baijiu samples in this article are anonymous, and Baijiu samples are only for research purposes without any commercial purpose.

### 2.3. Pretreatment Methods for Baijiu Samples

The Baijiu sample was shaken vigorously, left to stand for 10 min, filtered and purified with a 0.22 mm filter membrane (Nylon filter membrane, Beijing InnoChem Science & Technology Co., Ltd, Beijing, China), and then injected into the sample. The process was repeated three times for each sample to ensure accuracy and quality.

### 2.4. Analytical Parameters with GC-MS Detection

The instruments used in this research were Agilent Technologies 7890B GC System and Agilent Technologies 5977A MSD.

Each sample (1 μL) was injected in a splitless mode and analyzed on a DB-WAX column (60 m × 0.25 mm i.d., 0.25 μm film thickness, J&W Scientific, Folsom, CA, USA), respectively, for a cross-check of their RIs. Helium was used as the carrier gas at a constant flow rate of 1.5 mL/min. The injector temperature was 250 °C. The temperature program of the oven was as follows: the oven temperature was held at 40 °C at first, then raised to 200 °C at a rate of 30 °C/min and held for 2 min, then raised at 2 °C/min up to 240 °C and held for 10 min. The MS was operated in an electron ionization (EI) mode at 70 eV. The activation voltage was 1.5 V. The solvent delay was 4–8 min. The temperatures of the interface and the ion source were set at 250 and 230 °C, respectively. The identification of LCFAEEs was conducted in full scan mode. The mass range was set from 45 to 350 amu. The quantification of LCFAEEs was performed in selected-ion-scan mode (SIM).

### 2.5. Qualitative Analysis of LCFAEEs

Baijiu samples were subjected to full scan analysis following the analytical conditions described above. Qualitative analysis was performed by aligning the retention times of LCFAEEs in Baijiu with those of standards, combined with comparative analysis of standard mass spectra provided by the NIST 20 library according to matching degree and characteristic ions.

### 2.6. Quantitative Analysis of LCFAEEs

After fully shaking the Baijiu and letting it stand for 10 min, 1 μL of the Baijiu sample was injected into the GC in splitless mode and analyzed with the same conditions as described in Section 2.4 for the identification of LCFAEEs.

The detector temperature was set to 250 °C, and then stock solutions of mixed standards were prepared in absolute ethanol and diluted to a series of concentrations using 60° ethanol as the working standard solution. Afterwards, working standard solutions were analyzed by GC-MS (gas chromatography-mass spectrometer), the six LCFAEEs were linearly regressed with the mass concentration of the solution as the abscissa and the peak area as the ordinate, and the calibration curves were drawn at the same time. The concentrations of the LCFAEEs were calculated based on the calibration curves.

The analytical limits of detection (LOD) of LCFAEEs were obtained from the lowest concentrations of the analyte standard solutions based on a signal-to-noise ratio of 3. The limit of quantitation (LOQ) was based on a signal-to-noise ratio of 10. All analyses were repeated in triplicate.

The interday precision and intraday precision of six LCFAEEs at specific concentrations were calculated. The same mixture was tested at three different time periods during the same day, and the relative standard deviation of the contents of the six targets detected three times in a day was calculated to obtain the intraday precision and recovery rate. The same master mix was taken and tested by injection in the same period for five days, which lasted for five days. The relative standard deviations for the concentrations of the six LCFAEEs over five days were calculated to give interday precision.

### 2.7. Modern Flavor Sensomics Evaluation

The identification of the unique style and quality of Baijiu in this experiment was akin to the international sensory identification of food. It was realized by combining sensory, physical, and chemical experiments. For example, researchers conducted studies on Chi-aroma-type Baijiu and analyzed the effects of aged pork fat on the flavor during their aging. In addition, researchers also gave an overview of the synthesis of fatty acid ethyl esters and their flavor, which were found to have an impact on food flavor [31,32].

#### 2.7.1. Baijiu Evaluation Environment and Conditions

The environment was free of vibration and noise; the Baijiu evaluation room was clean and tidy, without a foreign smell; the air was fresh; and the temperature was 15–20 °C. The Baijiu tasting table was paved with a white tablecloth, a mouthwash cup, etc. The Baijiu sample temperature is 20 ± 1 °C, and the wine sample was numbered and grouped. The time of Baijiu evaluation was generally between nine and eleven o’clock or between fifteen and seventeen o’clock. Considering the referential nature of sensory experiment results, sensory evaluators with and without professional knowledge backgrounds were tried. Specifically, ten Baijiu sensory evaluators with relevant professional knowledge backgrounds (from the Beijing Key Laboratory of Flavor Chemistry, aged from twenty-three to twenty-seven) and ten Baijiu sensory evaluators without relevant professional knowledge backgrounds (from the Beijing Technology and Business University, aged from twenty-three to twenty-seven) were recruited to participate in a double-blind evaluation trial.

#### 2.7.2. Descriptive Profile Tests

A sensory evaluation for four dimensions were scored from: visual sensory, olfactory sensory, taste sensory, and evaluation after drinking, respectively (Table 3). The visual sensory scores used an eleven-point scale (i.e., 0–10, from extreme haze to clear-transparent); Olfactory sensation uses a six-point scale (i.e., 0–5, from none to very strong); an eleven-point scale (i.e., 0–10, from no aroma to extremely pungent stimuli-strong aroma, soft liquor taste); the evaluation after drinking (i.e., 0–25, from extreme dizziness to dizziness; and 0–10, from no retention in the mouth to noticeable retention in the mouth) finally combined the above four dimensions with the interval delineation of Baijiu quality.

### 2.8. Statistical Analysis

Origin 2018 (OriginLab Corporation, Northampton, MA, USA) software to draw radar diagrams and fingerprints; Tbtools (Beijing Liuzhi Information Technology Co., Ltd, Beijing, China) to draw heatmaps; SIMCA 14.1 software(Sartorius Stedim, Göttingen, Germany) to perform orthogonal partial least squares discriminate analysis (OPLS-DA) and predictor variable importance projection (VIP); SPSS 24.0 software(International Business Machines Corporation, Chicago, IL, USA) for single factor and correlation analysis; and R software(RStudio, Inc, Boston, MA, USA) is used to extract and visualize the results of principal component analysis (PCA) [33].

## 3. Results and Discussion

### 3.1. Parameter Description

Selected ion monitor (SIM) parameters used for qualitative and quantitative analysis of six long chain fatty acid ethyl ester (LCFAEEs) are exhibited in Table 4. In addition, linear regression was performed with the mass concentration of the solution as the abscissa and the peak area as the ordinate, and then the standard curve was plotted. The range of linearity, regression equations (Table 5), limits of detection (LOD), and limits of quantification (LOQ) of LCFAEEs were monitored and shown in Table 4. The intraday precisions of the six LCFAEEs were less than 5%, the interday precisions were less than 10%, and the recoveries ranged from 95 to 108% (Table 4). It was illustrated that the proposed method exhibited good accuracy and precision for detecting these six LCFAEEs and was fully capable of meeting the needs of analysis.

### 3.2. Analysis of the Distribution for Six LCFAEEs in Baijiu Produced Different Years

To reveal the influence mechanism of the production year on the distribution of LCFAEEs in Baijiu, the concentrations of LCFAEEs in the Baijiu sample groups (A1, A2, C1, C2, and C3-1) were measured. Overall, significant differences (*p* < 0.05) were observed between the total concentrations of LCFAEEs in Baijiu from different production years, ranging from 4.38 to 196.95 mg/L. The concentrations for EP (concentrations ranged from 1.14 to 100.75 mg/L), E9 (0.98–28.50 mg/L), and E912 (1.02–59.41 mg/L) in the six LCFAEEs were significantly higher than those for ET (0.30–3.21 mg/L), EO (0.94–2.42 mg/L), and E91215 (0.00–2.66 mg/L).

The concentrations of EP, E9, and E912 in the A1 and A2 Baijiu sample groups were significantly different (*p* < 0.01), whereas there was no significant difference between the concentrations of ET, EO, and E91215 (*p* > 0.05) (Figure 2a). In the comparison of the concentrations of six LCFAEEs in the C2 and C3-1 groups, there was a significant difference in EP concentration (*p* < 0.01), while no significant difference was found in the concentration of the other five LCFAEEs (P > 0.05) (Figure 2b). Compared with the C2 and C3-1 groups, C1 had significantly different concentrations of the six LCFAEEs (*p* < 0.01).

In comparison, it can be found that the concentration of Baijiu varied greatly between production years. Based on the variance analysis, it could be observed that EP had a remarkable difference in concentration among Baijiu groups of different production years. For the same level Baijiu groups, the concentrations of LCFAEEs in Baijiu produced in 2014 and 2015 were significantly higher than those in Baijiu produced in 2018 and 2022. It can be seen that Baijiu with increased storage time may increase the concentration of LCFAEEs in Baijiu; however, the exact reason remains to be further studied. In terms of physical changes, Baijiu undergoes volatilization and molecular association, which cause changes in the proportions of trace components such as alcohols, acids, esters, and so on in Baijiu. Chemical changes, on the other hand, include hydrolysis, esterification, and condensation of trace components. It has been observed that LCFAEEs in Baijiu change during storage, possibly due to processes of physical and chemical change [34,35].

### 3.3. Analysis of the Distribution of Six LCFAEEs in Different Grades of Baijiu

A1, B1, and C1 Baijiu sample groups produced in the same year were selected to analyze the concentration distribution of six LCFAEEs and the relationship between LCFAEEs and Baijiu grades. In general, the concentrations of six LCFAEEs in different grades of Baijiu were significantly different (*p* < 0.05). The concentration ranges of ET, EP, EO, E9, E912, and E91215 were: 0.27–2.31 mg/L, 0.75–47.41 mg/L, 0.93–1.80 mg/L, 0.98–12.87 mg/L, 1.01–27.08 mg/L, and 1.00–1.75 mg/L, respectively. The concentrations for each LCFAEE as well as their total concentrations exhibited the same trend in different grades of Baijiu samples. The highest total concentration of LCFAEEs was found in the C1 groups (93.23 mg/L), while the lowest (4.94 mg/L) was obtained from the B1 groups (Figure 3a).

Based on the concentration profiles of six LCFAEEs in different grades of Baijiu, further discriminant analysis was performed to resolve the effects of the concentration differences of the six LCFAEEs on Baijiu quality. Furthermore, effective differentiation for different levels of Baijiu samples could be achieved through OPLS-DA (Figure 3b). Meanwhile, after 200 times of permutation, the intersection points of the Q2 regression line and the vertical axis were less than zero, which indicated that there was no overfitting of the model and the model validation was valid, and this result was considered to be useful in the discrimination analysis for different levels of Baijiu samples (Figure 3c). It can be seen that the concentration profiles of six LCFAEEs in different grades of Baijiu significantly varied and were presumed to affect Baijiu quality. Of note, the concentrations of LCFAEEs in level 1 Baijiu samples were markedly higher than those in excellent and premium Baijiu samples (*p* < 0.01). Furthermore, the discriminatory power of six LCFAEEs was ranked for different grades of Baijiu. Based on the discriminant model, E91215, ET, and EP were confirmed as key factors for Baijiu grade discrimination due to the significant correlation between their concentrations and the corresponding Baijiu grade (Figure 3d).

### 3.4. Analysis of the Distribution of Six LCFAEEs in Baijiu under Different Sun Exposure Times

At different sun exposure times, Baijiu sample groups (C3-1, C3-2, C3-3, C3-4, C3-5, and C3-6) were selected to determine the change mechanism of LCFAEEs during the sun exposure process. Overall, significant differences (*p* < 0.05) were observed between the total concentrations of LCFAEEs in Baijiu at different sun exposure times. In addition, none of the concentrations for E91215 reached the limit of quantification, and significant differences (*p* < 0.05) were only found between the concentrations of ET and EP during the sun exposure process but not in the concentrations of EO, E9, and E912 (*p* > 0.05) (Figure 4a). This may be due to the fact that sun exposure accelerated the esterification of trace components in Baijiu, which transformed long-chain fatty acids into LCFAEEs. With increasing exposure time, LCFAEEs undergo hydrolysis or fission reactions. Compared with different production years and different levels of Baijiu samples, different sun exposure times mainly affected the ET and EP concentrations in Baijiu, and there was a tendency of first significantly increasing and then significantly decreasing (Figure 4b). The highest total concentrations of LCFAEEs (5.95 mg/L) were found in the Baijiu sun exposure to twelve days; the total concentrations of LCFAEEs in the Baijiu sun exposure to zero days were the lowest at 4.38 mg/L (Figure 4c).

To further explore the effects of the sun exposure process on LCFAEEs in Baijiu, a discriminant model was established to evaluate the degree of variation for LCFAEEs, and it was found that the sun exposure process had a significant effect on the concentration changes of ET, EP, and E912 (Figure 4d,e).

Based on the above studies, it can be observed that sun exposure affects the levels of LCFAEEs in Baijiu, exhibiting an oscillating pattern, i.e., increasing first, then decreasing. Therefore, the influence of sun exposure on Baijiu quality is worthy of further research. This provides a foundation for the optimization of the aging process of Baijiu. Additionally, the identification of key trace components in Baijiu could facilitate more stringent monitoring of Baijiu’s quality. Moreover, the findings of this experiment, which concur with previous studies that sun exposure can alter the quality of wine-like products, provide a practical basis for this purpose [36,37,38,39,40].

### 3.5. Modern Flavor Sensomics Analysis of Different Baijiu Samples

On the basis of the clear concentration profiles of LCFAEEs in different Baijiu sample groups, their contributions to Baijiu quality were further dissected. Subject groups were utilized to confirm the Baijiu samples flavor profile using modern flavor sensomics. Specifically, four dimensions (i.e., visual sense, olfactory sense, taste sense, and evaluation after drinking) were selected to exhibit the flavor profile of Baijiu samples. Based on this, the multivariate chemical analysis method was combined to further clarify the correspondence between LCFAEEs and the sensory dimension of Baijiu.

#### 3.5.1. Visual Sense

In a visual sense, whether the evaluators had relevant professional backgrounds did not significantly affect the scoring results (*p* > 0.05). The visual sensory scores of the A2 group were significantly lower than those of the other group samples (*p* < 0.01), and the Baijiu appeared slightly turbid (Figure 5). The reason for this phenomenon was presumed to be the excessive concentration of LCFAEEs in Baijiu in A2 groups [34,41].

#### 3.5.2. Olfactory Sense

For olfactory sense, whether the evaluators had relevant professional backgrounds was not found to be significantly different from the scoring results (*p* > 0.05). Significant differences (*p* < 0.05) in olfactory sensory scores were found between different Baijiu groups, which was consistent with the conclusions of other investigators. However, the correlation analysis between the concentrations of LCFAEEs in different baijiu groups and the olfactory sensory scores of Baijiu was not significant (*p* > 0.05). In general, multivariate analysis results showed that it was positively correlated with Baijiu aroma (grain aroma) and negatively correlated with the other four aroma attributes (Figure 6).

There are still many new ideas and methods in the related research of food senses that deserve our study. Such as the introduction of e-tongue, e-nose, etc. in future sensory studies of Baijiu. It was also considered necessary to establish a certain suitable model to predict the relationship between the concentrations of LCFAEEs and liking degree of Baijiu [42].

#### 3.5.3. Taste Sense

For taste, whether the evaluators had the relevant professional background produced significant differences in the scoring results (*p* < 0.05). However, the resulting differences were mainly reflected in the following: professional sensory evaluators were less sensitive to stimuli at the Baijiu entrance, and nonprofessional sensory evaluators are more sensitive to the intensity of a Baijiu as it releases its aroma in the mouth. Interestingly, the trend was the same between professional sensory evaluations and nonprofessional sensory evaluations for the overall taste sensory perception of Baijiu.

Based on multivariate analysis of the distribution of LCFAEEs in different Baijiu groups, the correlation between LCFAEE concentrations and taste perception scores was further evaluated. Specifically, the concentrations of LCFAEEs were significantly positively correlated with the inlet taste score (*p* < 0.05, Pearson correlation (*r*) = 0.937) and negatively correlated with the aroma sensation in the mouth score (*p* < 0.05, *r* = −0.745). Higher concentrations of LCFAEEs caused a remarkable decrease in the irritation sensation during the entrance and drinking process for Baijiu (Figure 7a,c); they caused a decrease in the aroma sensation in the mouth (Figure 7b). Based on this, it could be inferred that LCFAEEs contributed significantly to the “ softness “of Baijiu quality, and paradoxically, their increasing concentrations could lead to a weakening of the aroma intensity when the Baijiu was consumed. Therefore, for the concentration of LCFAEEs in Baijiu samples, a clever balance is required.

#### 3.5.4. Evaluation after Drinking

For the evaluation after drinking scores of Baijiu groups, whether the evaluators had relevant professional backgrounds had significant differences (*p* < 0.05) in the scoring results; the differences mainly manifested in the strong vertigo sensation after moderate drinking from nonprofessional sensory evaluations, and the scores were overall low. However, the same trend of post-beverage scores for Baijiu groups by professional sensory evaluations and nonprofessional sensory evaluations did not yield contradictory conclusions.

For the evaluation after drinking Baijiu, whether the evaluators had relevant professional backgrounds, did not produce significant differences in the scoring results (*p* > 0.05). The distribution of LCFAEEs in different Baijiu groups combined with modern flavor sensomics revealed that their concentrations were significantly correlated (*p* < 0.05, *r* = 0.931) with the retaining taste score (Figure 8a), but not with the drunkenness degree score (*p* > 0.05). However, there was a phenomenon worth noting: the higher-grade Baijiu and the longer the storage time, the lower the vertigo sensation after drinking (Figure 8b). It was suggested that the vertigo sensation after Baijiu drinking was the result of the interaction of multiple trace components in Baijiu, and the higher the quality of Baijiu, the lower the vertigo sensation after drinking.

#### 3.5.5. Comprehensive Analysis of Sensory Evaluation for Baijiu Samples

A discrimination model was conducted to further explore the comprehensive effect of LCFAEEs on different sensory dimensions of Baijiu. Consequently, these two dimensions (i.e., taste sense and evaluation after drinking) had a significantly higher influence on the sensory quality of Baijiu than the dimensions of visual sense and olfactory sense (Figure 9).

Combined with the above analysis, it could be found that LCFAEEs were one of the key factors for Baijiu quality, which could significantly improve the Baijiu’s “ softness “to some extent. However, the effect of single LCFAEEs on Baijiu quality has not been clarified. Next, single-factor sensory addition experiments will be conducted to clarify whether single LCFAEEs have significant effects on Baijiu sensory.

### 3.6. Baijiu Sample Addition Experiment

Based on the above findings, LCFAEEs have important effects on Baijiu flavor quality; furthermore, this study conducted Baijiu addition experiments to explore the effect of single LCFAEEs on Baijiu quality. To ensure the practicalities of recombination experiments, only the effects of LCFAEEs on the sensory quality of Baijiu were explored using the controlled variable method. Three different grades of Baijiu, A1, B1, and C1, were selected for additional experiments by our research group. Among them, the B1 Baijiu groups were chosen as the base Baijiu for additional experiments. Specifically, target concentrations for LCFAEEs were added to B1, which made its concentration match that of A1 or C1. The sensory evaluation of the recombinant Baijiu samples was performed three times and numbered as A1-6L, A1ET, A1EP, A1EO, A1E9, A1E912, A1E91215, C1-6L, C1ET, C1EP, C1EO, C1E9, C1E912, and C1E91215.

For the visual sense dimension, no significant (*p* > 0.05) difference was observed between the recombinant Baijiu samples and the original Baijiu samples. To explore the effects of LCFAEEs on the recombinant Baijiu samples in the olfactory sense dimension, OPLS-DA analysis was used to compare B1 Baijiu and recombinant samples simultaneously with different aroma attribute scores. It was found that effective differentiation could not be achieved for B1 and recombinant samples, and no significant differences (*p* < 0.05) in the scores of different aroma attributes were observed between the sensory evaluators scores for B1 and recombinant samples.

For the taste sensory dimension, two attributes, the aroma sensation in the mouth and laryngeal taste, were not significantly different (*p* < 0.05) between the scores of sensory evaluators for B1 Baijiu and recombinant samples. As for the inlet taste attribute, the scores of sensory evaluators for B1 Baijiu and recombinant samples were significantly different (*p* > 0.05), with LCFAEEs significantly reducing the irritation sensation at the entrance of Baijiu. Combined with the analysis in Section 3.5.3, it could be clarified that LCFAEEs could significantly improve Baijiu softness, among which the EP, E9, and E912 lift effects were more obvious (Figure 10a).

The addition of LCFAEEs did not significantly affect the drunkenness attribute for the recombinant samples (*p* > 0.05) and significantly affected the remaining taste attribute (*p* < 0.05), which were the same as the results in Section 3.5.4. In a word, LCFAEEs could significantly improve the remaining taste time of Baijiu, especially for EP and E912 (Figure 10b).

It should be noted that there was a significant negative correlation (*p* < 0.05, *r* = −0.745) between the aroma sensation in the mouth attribute score and the concentration of LCFAEEs in Section 3.5.3, which was not validated in additional experiments. (Note: the proposed LCFAEEs that are more capable of influencing Baijiu quality in this paragraph are based on the concentrations incorporated into the actual Baijiu).

Based on the above, LCFAEEs were the key effectors affecting Baijiu flavor. Notably, its synergistic or antagonistic effects with other trace components in Baijiu still need further exploration. The contribution of LCFAEE concentrations to human efficacy should also be considered if experimental conditions permit [43].

## 4. Conclusions

This study examined the distribution of LCFAEEs in Baijiu for different years, grades, and sun exposure times. It was observed that the concentration of LCFAEEs in Baijiu was greatly affected by the storage time and Baijiu grade, while the concentrations of ET and EP were mainly influenced by the sun exposure process. Moreover, modern flavor sensomics and multivariate chemometrics have demonstrated that LCFAEEs play a significant role in the “softness”, “remaining taste” and “aroma sensation in the mouth” attributes of Baijiu. In addition, it has been demonstrated that the longer the storage time of Baijiu, the better the quality; sensory personnel rated the Baijiu composite highest after 12 days of sun exposure. Thus, production processes that can effectively control the concentration of LCFAEEs in Baijiu may be the key to producing high-end products. Additionally, modernizing the Baijiu industry is essential for the stability of Baijiu quality. Sun-exposure can accelerate the aging process of Baijiu and reduce storage costs, offering a new idea for Baijiu production. By harnessing the effects of the sun, energy can be conserved and production costs can be cut. The results of the study suggested that LCFAEEs were not exclusive to the Baijiu industry. Additionally, whether LCFAEEs have potential effects on improving the flavor quality of other beverages is worth further investigation. By using LCFAEEs, it is possible to successfully adjust the balance of flavors, creating a smoother and more enjoyable beverage.

## Figures and Tables

**Figure 1 foods-12-01267-f001:**
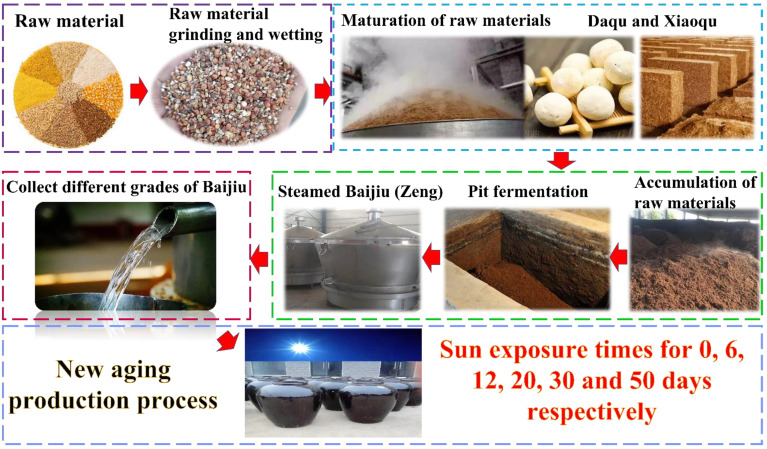
Overview of Baijiu production process and new aging process. (Notes: “Zeng: a container for producing liquor; “Daqu: a kind of fermentation and saccharification starters that is made of barley, wheat, peas, etc., and on which various microorganisms enrich naturally “; “Xiaoqu: a kind of fermentation and saccharification starters which is made of bran by artificial steaming, sterilization and inoculation of strains “).

**Figure 2 foods-12-01267-f002:**
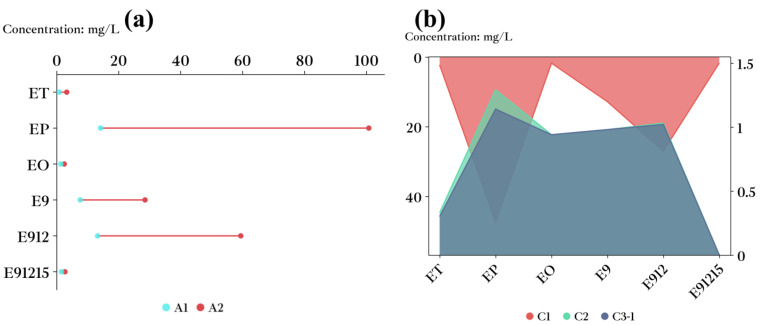
Concentration of six LCFAEEs for Baijiu in different production years. (**a**) Concentration of six LCFAEEs in A1 and A2 Baijiu. (**b**) Concentration of six LCFAEEs in C1, C2, and C3-1 Baijiu.

**Figure 3 foods-12-01267-f003:**
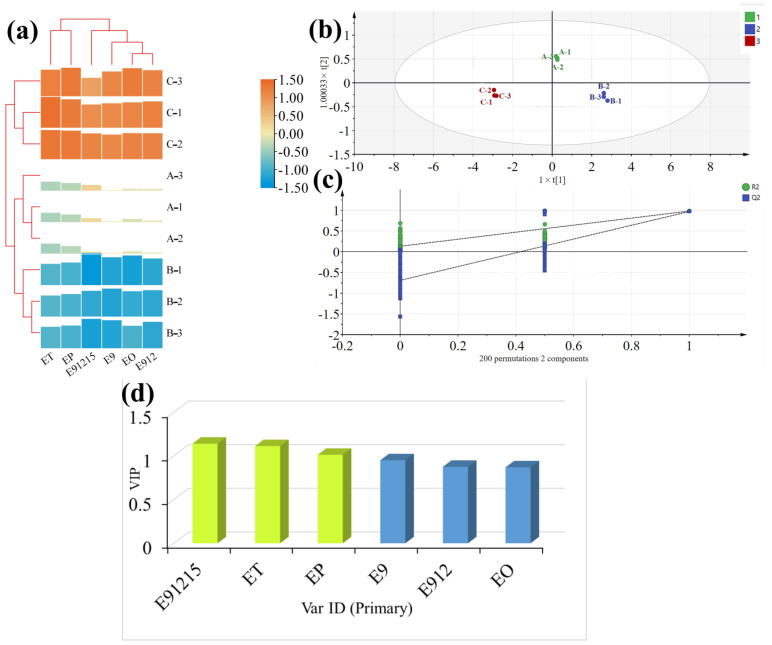
Analysis chart of six LCFAEEs from different grades of Baijiu. (**a**) A heatmap of six LCFAEEs from different grades of Baijiu. (**b**) OPLS-DA of the concentrations of six LCFAEEs from different grades of Baijiu. (**c**) Permutation test for the concentrations of six LCFAEEs from different grades of Baijiu. (**d**) VIP for the concentrations of six LCFAEEs from different grades of Baijiu.

**Figure 4 foods-12-01267-f004:**
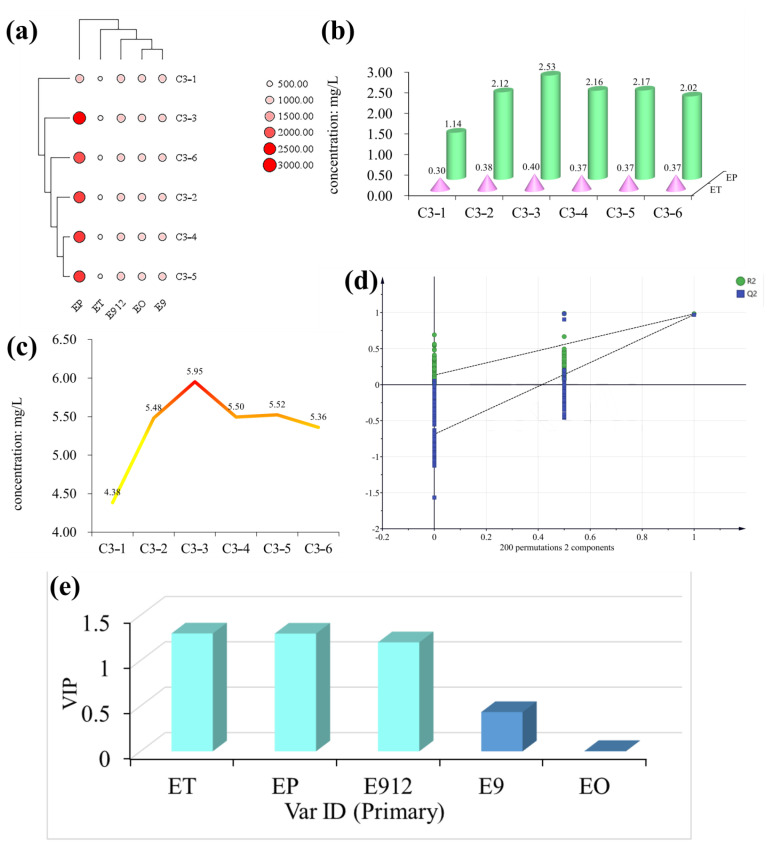
Dynamic changes of six LCFAEEs in Baijiu with different sun exposure times. (**a**) A heatmap of six LCFAEEs in Baijiu with different sun exposure times. (**b**) Change in the trend of ET and EP concentration in Baijiu with different sun exposure times. (**c**) The change trend of the total concentration of middle and LCFAEEs in Baijiu with different sun exposure times. (**d**) A permutation test of six LCFAEE concentrations in Baijiu with different sun exposure times. (**e**) VIP for the concentrations of six LCFAEEs in Baijiu with different sun exposure times.

**Figure 5 foods-12-01267-f005:**
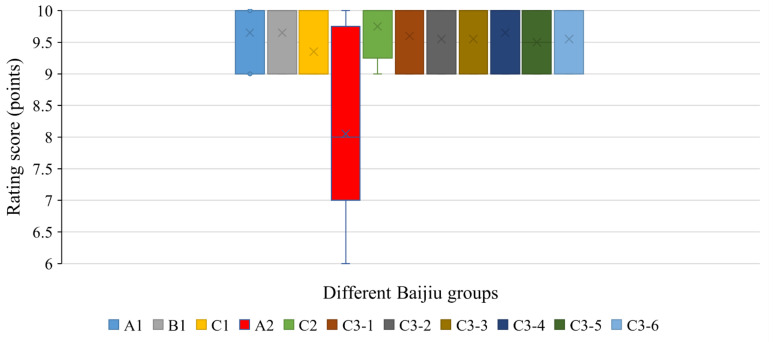
Visual sensory evaluation of different Baijiu groups.

**Figure 6 foods-12-01267-f006:**
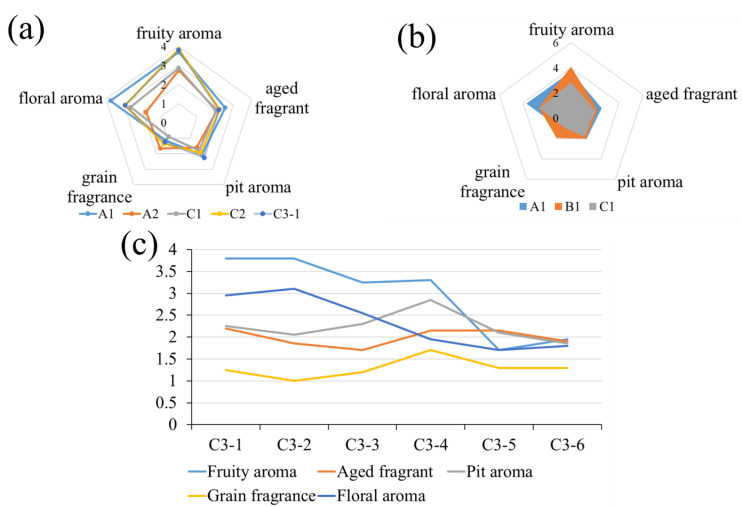
Olfactory sense evaluation of different Baijiu groups. (**a**) Olfactory sense evaluation of Baijiu in different production years. (**b**) Olfactory sense evaluation of different grades of Baijiu. (**c**) Olfactory sense evaluation of Baijiu with different sun exposure times.

**Figure 7 foods-12-01267-f007:**
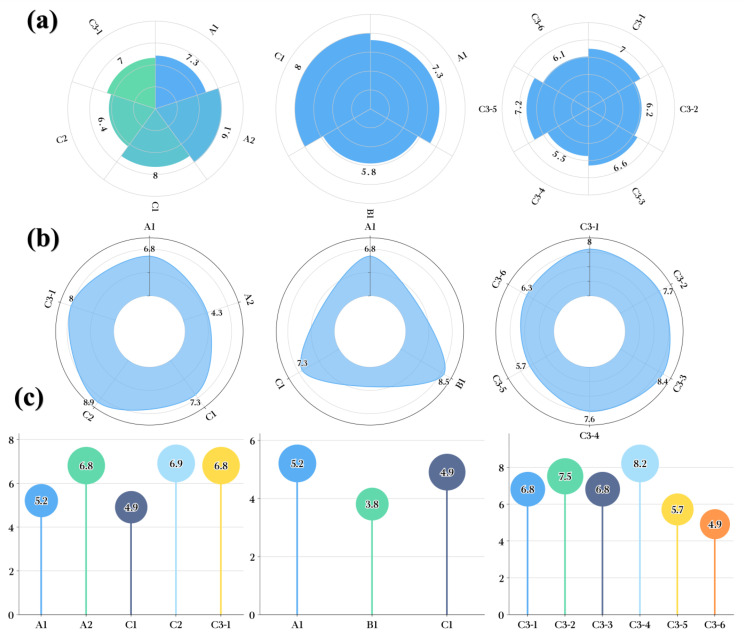
Taste and sensory evaluation of different Baijiu groups. (**a**) Inlet taste evaluation of Baijiu in different production years, different grades, and different sun exposure times. (**b**) A taste in the mouth evaluation of Baijiu in different production years, different grades, and different sun exposure times. (**c**) Middle laryngeal taste evaluation of Baijiu in different production years, different grades, and different sun exposure times.

**Figure 8 foods-12-01267-f008:**
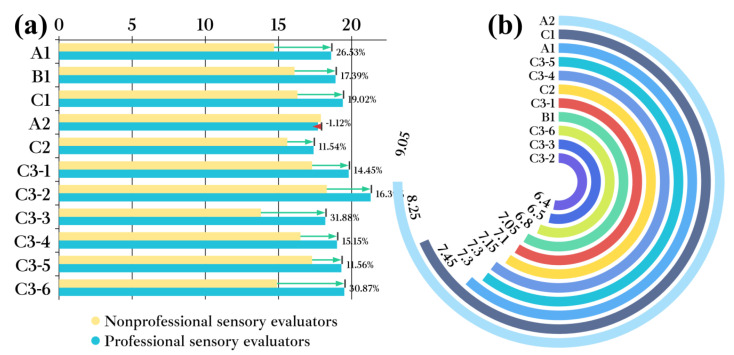
Evaluation after drinking evaluation of different Baijiu samples. (**a**) Evaluation of drunkenness degrees for different Baijiu samples after drinking. (**b**) Evaluation of the remaining taste for different baijiu samples.

**Figure 9 foods-12-01267-f009:**
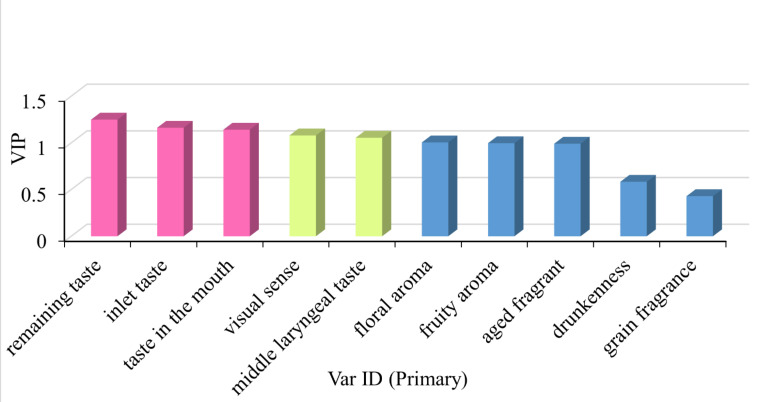
VIP for all sensory indicators of different Baijiu groups.

**Figure 10 foods-12-01267-f010:**
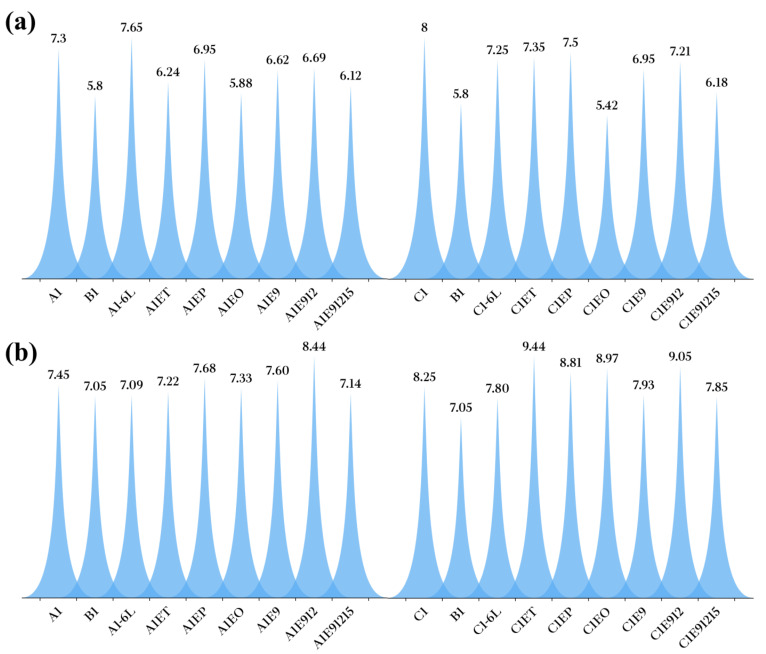
Sensory comparison between recombinant Baijiu samples and original Baijiu samples. (**a**) Sensory comparison between recombinant Baijiu samples and original Baijiu samples (inlet taste). (**b**) Sensory comparison between recombinant Baijiu samples and original Baijiu samples (remaining taste).

**Table 1 foods-12-01267-t001:** Standard information.

Name	Abbreviation of Chart Name	CAS	Source of Standard	Fineness
Ethyl tetradecanoate	ET	124-06-1	Shanghai Yuanye Biotechnology Co., Ltd., Shanghai, China	GC ≥ 98.0%
Ethyl palmitate	EP	628-97-7	Shanghai Yuanye Biotechnology Co., Ltd., Shanghai, China	GC ≥ 98.0%
Ethyl octadecanoate	EO	111-61-5	Tanmo Quality Inspection-Reference Material Center, Changzhou, China	GC ≥ 99.0%
Ethyl 9-octadecenoate	E9	111-62-6	Tanmo Quality Inspection-Reference Material Center, Changzhou, China	GC ≥ 98.5%
Ethyl 9,12-octadecadienoate	E912	544-35-4	Shanghai Aladdin Biochemical Technology Co., Ltd., Shanghai, China	GC ≥ 97.0%
Ethyl 9,12,15-octadecatrienoate	E91215	1191-41-9	Beijing North Weiye Measurement Technology Research Institute, Beijing, China	GC ≥ 98.0%

**Table 2 foods-12-01267-t002:** Baijiu sample information for the experiment.

Baijiu Sample Groups	Baijiu Information Description	Alcohol/°
A1	2014-premium	60
B1	2014-excellent	60
C1	2014-level 1	60
A2	2015-premium	60
C2	2018-level 1	60
C3-1	2022-level 1-exposure-0 days	60
C3-2	2022-level 1-exposure-6 days	60
C3-3	2022-level 1-exposure-12 days	60
C3-4	2022-level 1-exposure-20 days	60
C3-5	2022-level 1-exposure-30 days	60
C3-6	2022-level 1-exposure-50 days	60

**Table 3 foods-12-01267-t003:** Sensory evaluation report for Baijiu samples.

One-Level Indicators	Two-Level Indicators	Specific Description	Rating Score (Points)
Visual sense	Clear and transparent without impurities	Clear and transparent, with no suspension and no precipitate.	0–10
Olfactory sense	Fruity aroma	The aroma of Baijiu presents a fruit aroma (i.e., banana, apple, pineapple, etc.), and the fruit aroma smells comfortable.	0–5
	Aged fragrant	With an obvious special aroma except Baijiu, it can be described as oak barrel aroma.	0–5
	Pit aroma	Rich cellar aroma with natural earthy astringency.	0–5
	Grain fragrance	Comfortable aroma of cooked grain.	0–5
	Floral aroma	Sweet flower fragrance (i.e., rose, chrysanthemum, etc.).	0–5
Taste sense	Inlet taste	The complex trace components in Baijiu can cause irritation, but the time is short and the irritation is not strong.	0–10
	Aroma sensation in the mouth	The full-blown Baijiu aroma resembled a volcanic eruption and filled the mouth.	0–10
	Middle laryngeal taste	No irritant sensation to the throat, and Baijiu is soft.	0–10
Evaluation after drinking	Drunkenness	After drinking an appropriate amount of Baijiu (5–25 mL), you may get drunk within 30 min, but your body is not uncomfortable.	0–25
	Remaining taste	After swallowing Baijiu, the time is more than 15 s until the aroma disappears in the mouth.	0–10

**Table 4 foods-12-01267-t004:** Key parameters of LCFAEEs.

Compounds	Qualitative Ion	Quantitative Ion	LOD/(μg/L)	LOQ/(μg/L)	Interday Precision/(%)	Intraday Precision/(%)	Recovery Rate/(%)
ET	88, 101, 157, 211, 256	88, 256	0.24	0.76	1.1	2.57	95–103
EP	88, 101, 157, 239, 284	88, 284	0.32	1.91	1.15	1.73	105–108
EO	88, 101, 157, 269, 312	88, 312	0.11	1.13	2.11	3.79	95–101
E9	55, 180, 222, 264, 310	55, 310	0.18	1.18	1.13	1.15	96–106
E912	67, 81, 109, 263, 308	67, 308	0.24	3.32	2.14	8.62	98–107
E91215	79, 95, 108, 121, 306	79, 306	0.44	1.15	2.35	0.85	102–103

**Table 5 foods-12-01267-t005:** Regression equations for LCFAEES.

Compounds	Regression Equation	*R* ^2^
ET	y = 307753x − 80514	0.9989
EP	y = 308141x − 217577	0.9976
EO	y = 1000000x − 929800	0.9896
E9	y = 968421x − 943401	0.9958
E912	y = 1000000x − 1000000	0.9950
E91215	y = 1000000x − 1000000	0.9799

## Data Availability

The data are available from the corresponding author.

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
