# Peer review of "Revelation for the Influence Mechanism of Long-Chain Fatty Acid Ethyl Esters on the Baijiu Quality by Multicomponent Chemometrics Combined with Modern Flavor Sensomics"

_foods, 2023, doi:10.3390/foods12061267_

Round 1

Reviewer 1 Report

The manuscript “2248080” deals with the revelation for the influence mechanism of long-chain fatty acid ethyl esters on the Baijiu quality by multicomponent chemometrics combined with modern flavor sensomics. The topic is interesting and relevant, and minor changes are needed.

General Comments:

According to the authors, the aim of the work was to study the distribution of LCFAEEs for Baijiu 18 in different years, different grades and different sun exposure times. Foods journal provides an advanced forum for studies related to all aspects of food research, with major emphasis on the science of food. Therefore, it is a well developed and described study, however there are some important issues that need to be addressed:

1) The introduction needs some work to better set up the stage for the aims of the study. It will help the author to better explain their results.

3) The authors should clarify the study novelty, as well as described it clear in manuscript, since there are others studies approximately covering the same aim (with small alterations), as follow:

- The effect of aged pork fat on the quality and volatile compounds of Chi-aroma Baijiu, Food Sci Technol 43, 2023, https://doi.org/10.1590/fst.109922.

- Characterization of an Aspergillus niger for Efficient Fatty Acid Ethyl Ester Synthesis in Aqueous Phase and the Molecular Mechanism, Frontiers in Microbiology 12 (2022), doi: 10.3389/fmicb.2021.820380.

Author Response

Dear teacher, please see the attachment.

Reviewer 2 Report

I am very grateful you for the invitation to review the manuscript foods-2248080 by Wu and coauthors "Revelation for the Influence Mechanism of Long-Chain Fatty Acid Ethyl Esters on the Baijiu Quality by Multicomponent Chemometrics Combined with Modern Flavor Sensomics”. The work is interesting but needs adjustments to increase the quality of the material.

Comments:

- Abstract: The specification of Baijiu is not clear in the abstract (although it is presented later, in line 33) and is more relevant information.

- Lines 39-40: Authors should highlight complicated technology compared to other distilled liquors.

- Lines 52-54: The sentence is not clear. Please rewrite the text.

- Lines 64-65: Please describe the main changes (chemical and/or biochemical) that occur in this phase.

- Introduction: The traditional production process should be better described to facilitate the verification of changes in the suggested process.

- Introduction: Changes applied to the process are not discussed in depth.

- The aim of the work is not clearly presented. The authors must describe the aim in the abstract and introduction, facilitating the understanding of the general objective.

- Table 2: Please specify whether the alcohol content value was determined or derived from information provided by the product manufacturers.

- Lines 112-114: Please check the correct form of the sentence.

- Lines 136-137: Please check the correct form of the sentence.

- Lines 138-140: Please review the sentence.

- Lines 188-189: Please review the sentence. Some sentences are meaningless.

- Lines 194-200: The sentence is more appropriate in material and methods than as a result.

- Lines 227-228: Indicate the changes or probabilities that lead to the result. It is not enough just to describe as greater or lesser values.

- Lines 271-273: The exposure process and its influence are not described.

- Results and discussion: The citation of any article in this item is not verified. The authors present their results, without discussion with chemical, biochemical, or any transformation aspects. There is no comparison or use of previously published material.

- Results and discussion: Authors must add adequate discussion.

- Conclusion: It should be reviewed and include indications of related phenomena, but only exposure, time, etc.

Author Response

(The authors gave the same response as above.)

Reviewer 3 Report

The manuscript entitled Revelation for the Influence Mechanism of Long-Chain Fatty Acid Ethyl Esters on the Baijiu Quality by Multicomponent Chemometrics Combined with Modern Flavor Sensomics presented relevant results related to the evaluation of volatile compounds found in Baijiu and their effect on sensory evaluation. The manuscript is well-written but, has minor issues that authors must attend to prior to publication. Below are the comments.

-The abstract needs to be improved. For example, in lines 21-25 How much was increased/decreased to be significant? Include numerical values.

-Section 2.2. What was the control treatment?

-Line 113. Provide the details of the used membrane such as the material (nylon, PTFE, etc.) and the brand.

-Lines 146-147. Did the authors use an equation for estimating the signal-to-noise ratio? Include.

-Line 151. Change realized by done.

-Line 159-160. Homogenize using numbers for presenting the temperature values.

-Lines 211-214. Avoid using large digit amounts. Convert the amounts to mg/L. Revise the issue along the manuscript including the figures.

-Improve the quality and organization of figures 3 and 4. Figures 3b, 3c, and 4d have a watermark that says Trial. I recommend using licensed software to carry out the analysis.

Author Response

(The authors gave the same response as above.)

Reviewer 4 Report

The manuscript tittle: “Revelation for the influence mechanism of long-chain fatty acid ethyl esters on the baijiu quality by multicomponent chemometrics combined with modern flavor sensomics” investigates the influence mechanisms and distribution of long-chain fatty acid ethyl ester (LCFAEEs) on Baijiu quality. The manuscript is well written.

Some revisions should be done to improve the quality of the manuscript.

The aim of the work must be rewritten by clearly stating aim.

Abstract. Maybe, you should explain the LCFAEEs function or synthesis.

A brief description of material and method done in the study should be introduced. Or the experimental design. Baijiu was stored how many days?

Line 23. Level1 should be explained in the abstract.

Lines 24-25. It is not understandable why LCFAEEs increased with sun exposure, and after decreased. Is it due to the storage time?

Lines 26-28. The concentration of LCFAEEs in Baijiu were correlated with their flavor. Positively or negatively?

Introduction.

I would explain the elaboration process of Baijiu beverage in the text, although it has been explained in Figure 1.

How LCFAEEs is formed in the beverage?

You should introduce some references related to sensomics evaluation.

Material and method.

Line 104. Explain level1.

Lines 104. Explain premium and excellent.

Why samples were taken in different years?

Why 2022 samples were submitted to sun exposure?

You should explain the number of hours of sun exposure and how do you do this?

You should explain better the design of the experiment done.

How many repetitions did you use?

Results and discussion

Line 228. You should explain why the amount of LCFAEEs in Baijiu increase with the storage time? Some references should be introduced to discuss the results.

A general result should be included after beverage is submitted to different sun exposure.

For what type of industry could the findings found in this manuscript be used?

The results of the study have been explained correctly, by more discussion of the results compared to the literature is necessary. For example, you can discuss your results with those previously found in the literature. A lot of researchers have been discussing the effect of sensory evaluation in foods (e.g: Sánchez et al., 2021-2023 in table olives) or with the using of nose or tongue sensor (e.g: Martín-Vertedor et al., 2020). You should also compare your results with those previously found in the literature. These researchers have been also compared the sensory evaluation with the volatile compounds or even other compounds to try to stablish models or even with other ways to assay the volatile aroma compounds of the sample such as electronic devise. You should discuss the matter in the manuscript and look also for references with the same food.

Conclusion.

A general recommendation for industry should be included after the study done. How many days of storage do you recommend? How much sun exposure do you recommend? The better sensory quality recommended?

In short, what is the best scenario to transfer to the industrial sector to improve the quality of the final product?

Author Response

(The authors gave the same response as above.)

Round 2

Reviewer 2 Report

The quality of the manuscript is improved.

Author Response

Thank You I am deeply grateful for your help and encouragement. Your support has enabled me to perfect my article, making it even more scientific and rigorous.

Reviewer 3 Report

The manuscript has been improved. I only have minor comments.

-In the abstract. Provide the data for the results not, only the p-value.

-The authors rebutted: We utilized standard curves and calculated the limits of detection and quantification. 

So, what was the equation for calculating the signal-to-noise ratio? Include in the manuscript.

Author Response

(The authors gave the same response as above.)

Reviewer 4 Report

The authors have made a number of changes to the text that significantly improve the quality of the publication. The language used in the article has been improved. Thoughts and facts were exposed clearly. The discussion part has been extended to discuss the results. The well-written manuscript about has a significant relevance to food science research field. Therefore, I believe that all necessary corrections have been made by the authors. I have no comments on the current content of the article. 

Author Response

(The authors gave the same response as above.)
